# On the scaling of polynomial features for representation matching

**Siddhartha Brahma**
IBM Research, Almaden, San Jose, USA
`brahma@us.ibm.com`

## Abstract

In many neural models, new features as polynomial functions of existing ones are used to augment representations. Using the natural language inference task as an example, we investigate the use of scaled polynomials of degree 2 and above as matching features. We find that scaling degree 2 features has the highest impact on performance, reducing classification error by 5% in the best models.

## 1 Introduction

In many tasks in natural language processing, it is necessary to match or compare two distributed representations. These representations may refer to whole sentences, word contexts or any other construct. For concreteness, let $u$ and $v$ be two representations we want to match. In order to facilitate the matching, it is often beneficial to explicitly create new features like element-wise absolute difference ($|u-v|$) and element-wise product ($u \cdot v$) that augment $u$ and $v$. The combined feature vector is then processed by further layers in the task specific neural network. For example, Tai et al. (2015) use these heuristics to improve semantic representations. Most notably, for the natural language inference task, augmenting the hypothesis ($u$) and premise ($v$) representations with $|u-v|$ and $u \cdot v$ considerably improves performance in a siamese architecture Mou et al. (2016). This is also used in the more sophisticated models of Chen et al. (2017), where $u$ and $v$ represent word contexts. Several of these approaches are explored in the Compare-and-Aggregate framework by Wang & Jiang (2017).

In this paper we focus on polynomial features like $u \cdot v$ for the natural language inference task, where it is trying to capture similarity between $u$ and $v$. It is also a monomial of degree 2. We investigate two aspects of such terms - the use of scaling and the use of higher degree polynomials. The motivation for the former is the following. The values taken by individual elements of $u$ and $u \cdot v$ will in general have slightly different statistical distributions. For example, if elements of both $u$ and $v$ are approximately zero mean Gaussians with variance $\sigma^2$, the variance of the elements of $u \cdot v$ will be approximately $\sigma^4$. As such, subsequent layers in the neural network use weights that are initialized assuming identically distributed inputs Glorot & Bengio (2010), which is clearly not the case when $\sigma \neq 1$. An appropriate scaling coefficient attached to $u \cdot v$ that can bring its variance close to that of $u$ is one possible way of addressing this anomaly. The motivation for the latter is to incorporate more complex multiplicative interaction between $u$ and $v$ through degree 3 and 4 polynomials.

Our findings are two-fold. Through numerical experiments using the Stanford Natural Language Inference (SNLI) Bowman et al. (2015) dataset, we show that in the absence of scaling, using higher degree polynomial features instead of $u \cdot v$ improves the performance of baseline models. In the presence of scaling, this difference all but vanishes and in fact the scaled $u \cdot v$ achieves the best performance. The use of scaling significantly reduces classification error, by up to 5.0% in the best performing models. This is observed both for models that only use encodings of whole sentences and more complex ones.

## 2 Scaled polynomial features

We present our work in the context of two baseline models for the natural language inference task. In this task, given a pair of sentences (premise and hypothesis), it needs to be classified into one of

three categories - entailment, contradiction and neutral. In the first model, both the premise and the hypothesis sentence are encoded using a bidirectional LSTM Hochreiter & Schmidhuber (1997) and the intermediate states are max-pooled to get the respective representations $u$ and $v$. We refer to this as the InferSent model Conneau et al. (2017). The standard matching feature of Mou et al. (2016) uses a concatenation of $u$, $v$, $|u-v|$ and $u \cdot v$. We define the following new matching feature vector that scales the multiplicative term by a constant factor $\eta > 0$.

$$\mathbf{w}_{poly2} = [u, v, |u - v|, \eta(u \cdot v)] \tag{1}$$

To incorporate polynomial multiplicative features between $u$ and $v$ of degree 3 and 4, we define the following define the following matching feature vectors.

$$\mathbf{w}_{poly3} = [u, v, |u - v|, \eta(u \cdot v) + \eta^2(u \cdot u \cdot v + u \cdot v \cdot v)] \tag{2}$$

and

$$\mathbf{w}_{poly4} = [u, v, |u-v|, \eta(u \cdot v) + \eta^2(u \cdot u \cdot v + u \cdot v \cdot v) + \eta^3(u \cdot u \cdot u \cdot v + u \cdot u \cdot v \cdot v + u \cdot v \cdot v \cdot v)] \tag{3}$$

In $\mathbf{w}_{poly3}$, the additional term is the sum of the two possible monomials of degree 3 involving both $u$ and $v$. In $\mathbf{w}_{poly4}$, the fourth degree term is the sum of the 3 possible monomials of degree 4 involving both $u$ and $v$. Note that we scale the 3rd degree terms by $\eta^2$ and the 4th degree terms by $\eta^3$. If the dimension of $u$ and $v$ is $d$, the dimensions of $\mathbf{w}_{poly2}$, $\mathbf{w}_{poly3}$ and $\mathbf{w}_{poly4}$ are all $4d$. In each case, the feature vector is fed into a fully connected layer(s), before computing the 3-way softmax in the classification layer. It is possible to use each of the degree 3 and 4 terms separately as a feature, but this did not make our models substantially more accurate. Choosing $\eta = 1$ in $\mathbf{w}_{poly2}$ reduces it to the matching feature vector proposed by Mou et al. (2016).

The same procedure is repeated for the other baseline model, namely ESIM Chen et al. (2017). In this case, $u$ represents one of the intermediate states of a bidirectional LSTM encoding of the premise (hypothesis) and $v$ represents the hypothesis (premise) states weighted by relevance to the premise (hypothesis) state. We replace the matching feature vector used in ESIM by the ones defined above. The rest of the model is the same which includes another bidirectional LSTM layer followed by pooling, a fully connected layer and a classification layer.

## 3    TRAINING AND RESULTS

For the InferSent model, we train on the SNLI dataset for the sentence encoding dimensions $d \in \{512, 1024, 2048, 4096\}$. The fully connected layer after $\mathbf{w}_{poly2}$, $\mathbf{w}_{poly3}$, $\mathbf{w}_{poly4}$ has two layers of 512 dimensions each. For optimization, we use SGD with an initial learning rate of 0.1 which is decayed by 0.99 after every epoch or by 0.2 if there is a drop in the validation accuracy. Gradients are clipped to a maximum norm of 5.0. Each experiment is repeated 5 times with random weight initializations and the average classification accuracies on the test set are reported. For training the ESIM model, we follow the procedure outlined in Chen et al. (2017) with the bidirectional LSTM dimension being $d = 600$.

Fig. 1(a) shows the dependence of the test accuracies with changing $\eta$ for each value of $d$ and matching feature vector $\mathbf{w}_{poly2}$ for InferSent. The variation of accuracy on $\eta$ is clearly visible and the best accuracy is obtained for $\eta = 32, 32, 16, 32$ for $d = 4096, 2048, 1024, 512$, respectively. This validates our intuition that the second degree feature ($u \cdot v$) should be scaled differently than the remaining first degree features in $\mathbf{w}_{poly2}$. For $d = 4096$, the average accuracy for $\eta = 32$ is 84.82%, which is 0.44% higher than that of $\eta = 1$. Comparing the best performing models for the different weight initializations, the one for $\eta = 32$ has almost 5% less error than $\eta = 1$.

Fig. 1(b) shows the same phenomenon for $\mathbf{w}_{poly3}$. The highest accuracies are obtained for $\eta = 4$ for $d = 512, 1024$ and for $\eta = 16$ for $d = 2048, 4096$. For $d = 4096$, the average accuracy for $\eta = 16$ is 84.73%, which is 0.15% higher than that of $\eta = 1$. Note that the performance tends to drop significantly for $\eta = 32, 64$, which points to possibly unstable training because of the large values of the coefficients $\eta^2$. We observe a similar trend for $\mathbf{w}_{poly4}$.

Fig. 2(a) compares the test accuracies of $\mathbf{w}_{poly2}$, $\mathbf{w}_{poly3}$ and $\mathbf{w}_{poly4}$ for $d = 4096$. Interestingly, the use of higher degree terms helps the classifier for $\eta = 1$, with $\mathbf{w}_{poly4}$ achieving 0.25% better accuracy than $\mathbf{w}_{poly2}$. However, with scaling $\mathbf{w}_{poly2}$ does progressively better while the other two models eventually suffer from unstable training. The overall highest average accuracy of 84.82% is

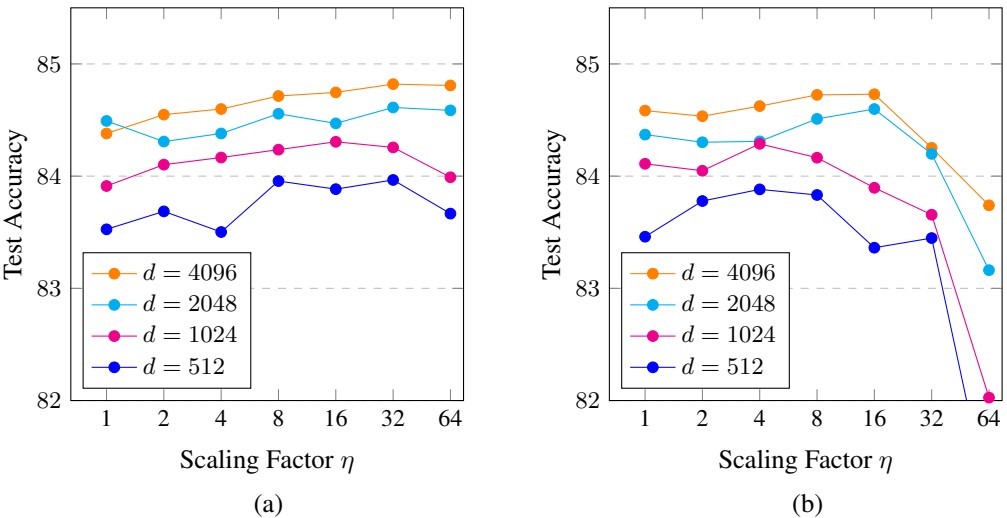

Figure 1: Average test accuracy of InferSent on the SNLI dataset for (a) $\mathbf{w}_{poly2}$ and (b) $\mathbf{w}_{poly3}$.

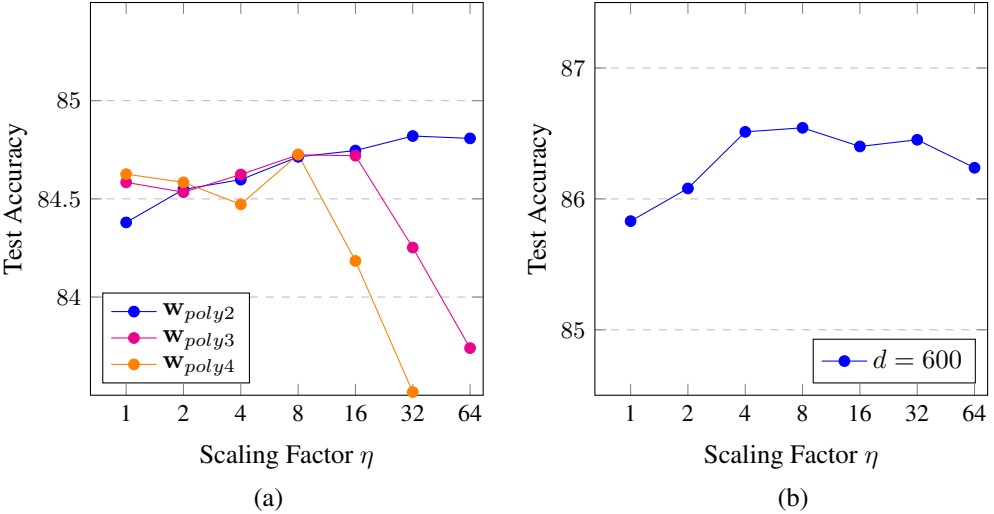

Figure 2: (a) Average test accuracy for $d = 4096$. (b) Test accuracy of $\mathbf{w}_{poly2}$ on the ESIM model.

achieved by $\mathbf{w}_{poly2}$ at $\eta = 32$ and the highest individual accuracy of 85.22% is achieved at $\eta = 16$ by the same model.

Finally, we report the test accuracy of $\mathbf{w}_{poly2}$ on the ESIM model for natural language inference. Here again, the gains for using scaled features is quite prominent, with a relative reduction in error of 5% for $\eta = 8$ as compared to $\eta = 1$.

## 4 CONCLUSION AND FUTURE DIRECTIONS

In this paper, we explore the use of higher degree polynomial features and scaling to derive new features when two distributed representations have to be matched or compared. Using the natural language inference task as an example, we show that scaling the higher degree terms helps reduce classification error, in some cases by almost 5%. In our preliminary experiments, we use constant scaling factors, but they can be learnt as a parameter of the model. We scale the exponent of $\eta$ as the degree of the monomials, which itself may be optimized to stabilize training for higher degree terms. Finally, it will be interesting to use the same scaling mechanism for tasks other than natural language inference.

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
