# OpenReview forum: "On the Scaling of Polynomial Features for Representation Matching"
_ICLR.cc/2018/Workshop — Reject_

### Official Review · AnonReviewer2 · 2018-03-04
**Lack of novelty and insightful analysis**

**Rating:** 3
**Confidence:** 4

**Review:**

The author investigates the effectiveness of using polynomial features to match sentences for the SNLI task.

The paper is clearly written. However, the study of polynomial features is decidedly not novel. Other than this, the analysis mostly show results of hyperparameter tuning without interpretation. For example, in which situations does the proposed method outperform the baseline? My other concern, in addition to the lack of novelty and insightful analysis, is that the author performs analysis using the test set. When the author says "validation accuracy", does this also refer to test set performance? If so, the performance of the model is consequently not comparable to existing results such as the ESIM model shown in Figure 2b.

Due to the lack of novelty and unconvincing experimental setup, I do not recommend the paper for publication.

---

### Official Review · AnonReviewer3 · 2018-03-09
**Novelty is limited and improvement needs more explanation**

**Rating:** 4
**Confidence:** 4

**Review:**

The paper explores scaling and higher degrees (i.e., 3 and 4) of polynomial features (i.e., u*v) for representation matching, tested on a natural language inference task. Two types of models, i.e., siamese architectures and ESIM (Chen et al. 2017), are compared in the experiments, which show that scaling helps and the higher-degree features are only useful without scaling (in Figure 2(a)), on the given baselines.

Cons:
1. The baseline model ESIM is reported to achieve an accuracy of 88.0% in (Chen et al. 2017), but in this paper, the performance of ESIM is reported to be 86.0%, which needs clarification.
2. The scaling method is pretty heuristic. The intuition is to relieve difference of variances, i.e., \sigma^2 and \sigma^4. Possible alternatives include using sqrt (u * v) not  \eta (u * v). The authors may consider having more discussion on this.

Pros:
With the given baselines, a relatively simple scaling method on features is shown to help representation matching. Polynomial features with more degree helps in some special cases.

---

### Official Review · AnonReviewer1 · 2018-03-10
**Interesting matching features with minor improvements**

**Rating:** 6
**Confidence:** 3

**Review:**

This paper presents an investigation on different scaling of polynomial features for semantic similarity scores. Their experimental results demonstrated that using the right scaling factor can improve natural language inference task.
In sum, this paper is clearly written. The author presents a minor focus contribution with some minor improvements.
To verify the importance of the polynomial features, I suggest to include also results without using the polynomial features (e.g. only [u, v, |u-v|]).
Furthermore, I suggest the author to perform an error analysis to have a better understanding in which cases the system was improved and why.

---

### Decision · Program_Chairs · 2018-03-20
**ICLR 2018 Workshop Acceptance Decision**

**Decision:**

Reject

**Comment:**

Based on the reviews, this paper has not been accepted for presentation at the ICLR workshop. However, the conversation and updates can continue to appear here on OpenReview.